# A Rare Case of Undifferentiated Pleomorphic Cardiac Sarcoma with Inflammatory Pattern

**DOI:** 10.3390/medicina58081009

**Published:** 2022-07-28

**Authors:** Alexandra Iulia Stoica, Marius Mihai Harpa, Cosmin Marian Banceu, Claudiu Ghiragosian, Carmen Elena Opris, Hussam Al-Hussein, Hamida Al-Hussein, Sanziana Flamind Oltean, Tibor Mezei, Razvan Gheorghita Mares, Horatiu Suciu

**Affiliations:** 1Emergency Institute for Cardiovascular Diseases and Transplantation Targu Mures, 540136 Mures, Romania; alexandra.stoica92@yahoo.com (A.I.S.); banceu.cosmin@yahoo.com (C.M.B.); ghiragosian_claudiu@yahoo.com (C.G.); carmenchincisan@yahoo.com (C.E.O.); alhussein.hussam@gmail.com (H.A.-H.); hamida.alhussein@yahoo.com (H.A.-H.); flamand.sanzi@yahoo.com (S.F.O.); tmezei@pathologia.ro (T.M.); razvan_mares_7@yahoo.com (R.G.M.); horisuciu@gmail.com (H.S.); 2Department of Surgery, George Emil Palade University of Medicine, Pharmacy, Science and Technology of Targu Mures, 540142 Mures, Romania

**Keywords:** cardiac tumors, undifferentiated pleomorphic sarcoma, cardiac surgery, cardiac malignancy, inflammatory response

## Abstract

Cardiac undifferentiated pleomorphic sarcoma (UPS) associated with fever and inflammatory response is an extremely rare condition. Herein, we report a rare case of cardiac UPS with unusual clinical presentation and inflammatory response. A 67-year-old male complaining of progressive dyspnea and intermittent fever of unknown cause was referred to our hospital for surgical resection of a left atrial mass. Laboratory analysis showed leukocytosis (26 × 10^3^/μL) and high C-reactive protein (CRP) levels (155.4 mg/L). Hemoculture tests and urine analysis were negative for infection. A contrast chest computed tomography revealed a mass measuring 5.5 × 4.5 cm, occupying the left atrium cavity. The patient underwent surgical excision of the mass, however, surgical margin of the resected tumor could not be evaluated, due to the multifragmented nature of the resection specimen. Postoperative CRP and leukocyte levels normalized, highlighting the relationship between the tumor and the inflammatory status. Early diagnosis is crucial for a proper management and favorable outcome, enabling patients to undergo chemotherapy and achieve complete surgical resection.

## 1. Background

Cardiac sarcomas are rare clinical entities, with high metastatic potential and aggressive clinical behavior associated with a poor survival rate [1].

Cardiac undifferentiated pleomorphic sarcoma (UPS) associated with fever and inflammatory response is an extremely rare condition [2].

Given the rarity of cardiac UPS, the locally advanced tumor at the time of diagnosis as well as the nonspecific clinical presentation, available data about cardiac UPS are limited.

Cardiac UPS typically develop in the left atrium (LA) and invade the atrial and ventricular walls with an infiltrative growth pattern, resulting in mitral valve dysfunction and subsequent left-sided heart failure [3].

The efficacy of current treatment is unclear. The protocol includes complete surgical resection if possible, followed by adjuvant chemotherapy and/or radiation therapy, to reduce the risk of recurrence [4].

This study reports a rare case of cardiac undifferentiated pleomorphic sarcoma with unusual clinical presentation and inflammatory response.

## 2. Case Report

### 2.1. Patient Information

A 67-year-old male patient with a history of progressively worsening dyspnea, fatigue, and intermittent fever of unknown cause, was late referred to a cardiologist.

The patient had a previous history of atrial fibrillation, stroke, and chronic kidney disease.

### 2.2. Preoperative Findings

The physical examination was notable for a systo-diastolic murmur throughout entire precordium. Electrocardiogram showed atrial fibrillation. Laboratory analysis showed leukocytosis (26 × 10^3^/μL) with a neutrophil-lymphocyte ratio of 20 and high C-reactive protein (CRP) levels (155.4 mg/L), and a normal value of procalcitonin (<0.5 ng/mL). Large spectrum antibiotic therapy was initiated as bacterial infection was suspected, however, the patient’s clinical condition did not improve.

Chest X-ray showed a dilated LA and excluded lung consolidations.

Hemoculture tests and urine analysis were negative for infection.

Transthoracic echocardiography (Figure 1) revealed a left atrial mass attached to the postero-lateral wall with an area of 18 cm^2^, severe secondary mitral stenosis, with a trans-valvular gradient of 27/12 mmHg, severe tricuspid regurgitation, moderate pulmonary hypertension, and a preserved left ventricular function with an ejection fraction of 70%.

A contrast total-body computed tomography (CT) revealed a mass measuring 5.5 × 4.5 cm, occupying the left atrium cavity (Figure 2 and Figure 3), as well as small hypodense lesions in brain. The atrial mass was interpreted as atrial myxoma, and the patient was referred for surgical treatment. Preoperative cardiac catheterization revealed normal coronary arteries.

### 2.3. Intraoperative Findings

The patient underwent surgical excision of the left atrial mass. During inspection, a large lobulated mass with a smooth surface and malignant aspect was found arising from the infero-posterior mitral annulus and extending to the posterior mitral leaflet. Wide surgical excision and mitral valve replacement were performed. Moreover, due to severe tricuspid regurgitation, tricuspid annuloplasty was performed as well. The tumor was sent for diagnosis confirmation at the pathology department.

### 2.4. Histological Findings

The histological diagnosis was high-grade undifferentiated pleomorphic sarcoma.

Microscopically, the tumor was composed mainly of atypical fusiform cells with a storiform pattern and rare epithelioid cells, areas of myxoid differentiation and nuclear atypia (Figure 4 and Figure 5). Less than 9 mitoses/10 HPF were identified, with less than 50% necrotic areas. According to the French Federation of Cancer Centers Sarcoma Group/FNCLCC, the histological findings correspond to score 5.

According to the American Joint Committee on Cancer (AJCC) and International Union Against Cancer (UICC) criteria, the tumor has been assigned a histological grade 2 (high grade).

Immunohistochemical (IHC) examination showed vimentin and smooth muscle actin positivity.

The tumor cells were negative for desmin, melan-A cytokeratin A1/AE3, CD34, HMB45, S100. The Ki-67 proliferation index ranged between 50 and 60%.

The surgical margin of the resected tumor could not be evaluated due to the multifragmented nature of the resection specimen.

### 2.5. Postoperative Evolution

Postoperative transthoracic echocardiography revealed a normal function of biological mitral prosthesis, mild tricuspid regurgitation and no further visualization of the mass.

The patient recovery was uneventful, except for a pericardial effusion which was surgically drained. Postoperative laboratory tests showed normal leucocytes (6 × 10^3^) and CRP level of 3.41 mg/dL. 

The patient was discharged and referred to the oncology department for further treatment.

Four months after surgical treatment, echocardiography reevaluation revealed recurrence of the mass in the left atrium. Due to the extent of the disease as well as the deterioration of the neurological status of the patient, further surgical excision was not possible, and the patient died after a few weeks.

## 3. Discussion

Primary cardiac tumors are extremely rare, the vast majority being benign, with only a quarter representing cardiac malignancies [5].

The presence of cardiac undifferentiated pleomorphic sarcoma (UPS) associated with neoplastic fever and leukocytosis is not extensively reported in the literature.

Previous studies have demonstrated that fever related to subsequent cancer occurs in approximately 2.3% of patients, and only a small proportion of these patients were diagnosed with sarcoma [6].

Neoplastic fever, which refers to fever caused by the malignancy itself, is extremely rare in patients with sarcomas. The diagnosis of neoplastic fever can be a dilemma in patients with cancer, and it should be considered after exclusion of other causes of fever, such as infection, leukemia or autoimmune disease [7,8].

The recognition of cardiac sarcomas can be difficult. Clinical presentation is nonspecific and may include symptoms related to obstruction, such as progressive dyspnea, pulmonary edema, chest pain, or embolic phenomena [9,10].

Due to delayed diagnosis and therapeutic difficulty, the management of sarcomas represent a challenge, and the long-term outcome is very poor, with median survival of less than one year [11].

The reported patient presented a mass that filled approximately 75% of the left atrium, and most of the symptoms were related to valve obstruction, except for fever.

Some studies suggested that neoplastic fever is associated with a more advanced disease and worse survival outcome [8]. In a study that analyzed the relationship between UPS and peripheral blood markers of systematic inflammation, such as neutrophil–lymphocyte ratio (NLR), a significant association was found between elevated NLR and a more advanced disease, high grade tumors, distant metastasis and poor prognosis [12].

The inflammatory status, such as high CRP levels, is associated with decreased survival rates in patients with many types of malignancies [13,14,15,16]. In a study conducted by Chan J et al. that assessed elevated CRP levels in patients with sarcomas, high CRP levels were associated with high grade tumors and poor prognosis [17]. In the presented case, after surgical excision, postoperative CRP and leukocyte levels normalized, highlighting the relationship between the tumor and the inflammatory status.

Surgical excision is considered the gold standard treatment for sarcomas. Complete surgical resection is associated with improved survival. However, achieving negative margins is difficult, due to the delayed diagnosis and extent of myocardial involvement

Partial resection of the tumor may have an impact on symptom relief and quality of life; nevertheless, it is associated with poor prognosis, especially in patients with neoplastic fever and inflammatory response [18,19]. Similarly, our reported patient experienced fever associated with an inflammatory response and presented a poor prognosis, with recurrence of the mass four months after surgical excision.

The utility of chemotherapy and radiation in UPS remains undefined.

Given the rarity of these tumors and lack of clinical trials, treatment modalities are extrapolative and chemotherapy regimen is derived from extra-cardiac sarcomas [20,21,22].

Combined neo-adjuvant chemotherapy and radiation may improve the result of surgical resection and reduce the risk of mass recurrence [23,24].

Several studies have shown that neo-adjuvant chemotherapy is associated with significant improvement in the rate of negative margins and overall survival [25,26].

In this case, surgical resection with negative margins was not possible due to the infiltrative pattern of the mass and delayed diagnosis. Chemotherapy and radiation treatment were palliative due to the presence of metastases at the time of diagnosis.

## 4. Conclusions

Cardiac sarcomas are an extremely rare condition and are associated with poor prognosis. Due to the difficulty in obtaining negative margins, survival after surgical resection is limited. Therefore, neo-adjuvant chemotherapy should be considered to aid surgical treatment.

Consequently, early diagnosis is crucial for a proper management and favorable outcome, enabling patients to undergo chemotherapy and achieve complete surgical resection.

In summary, we presented an atypical presentation of a rare UPS localization with inflammatory response and recurrent fever, which was associated with a poor prognosis.

## Figures and Tables

**Figure 1 medicina-58-01009-f001:**
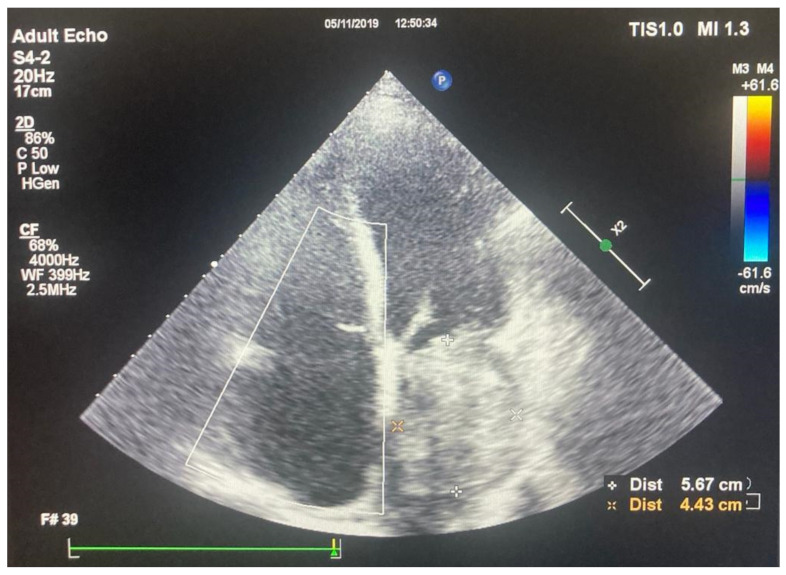
Transthoracic echocardiography, apical four chamber view showing a giant left atrial mass.

**Figure 2 medicina-58-01009-f002:**
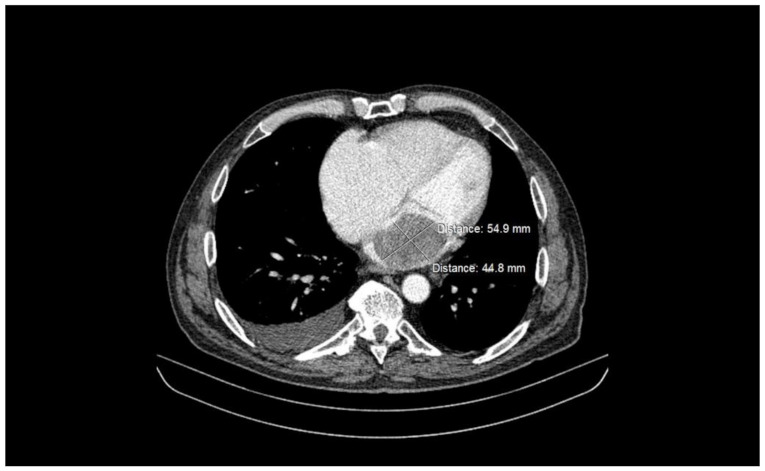
Total-body CT scan—arterial phase, axial view—intracavitary mass at the level of the left atrium, occupying most of the atrial cavity. The CT scan demonstrated a well-delimited, heterogenous mass attached to the lateral atrial wall, with no extension at the level of the mitral valve of the left ventricle.

**Figure 3 medicina-58-01009-f003:**
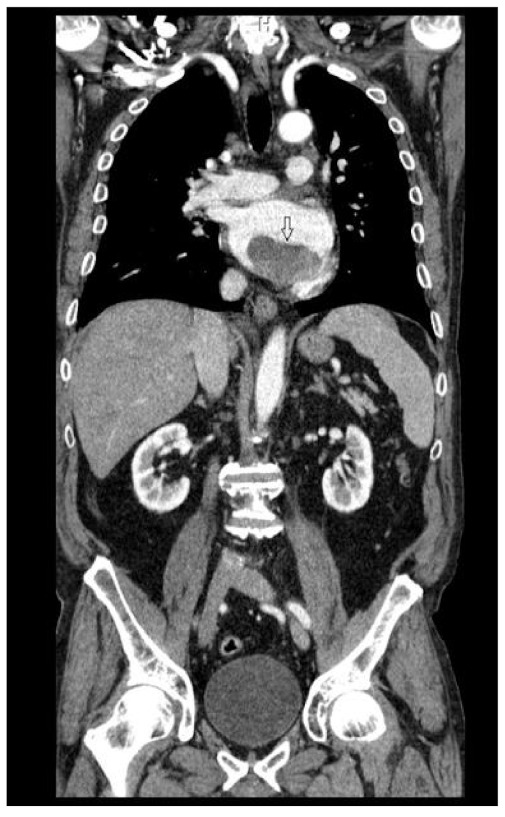
Total-body CT scan—arterial phase, coronal view—demonstrating the mass being attached to the inferior atrial wall.

**Figure 4 medicina-58-01009-f004:**
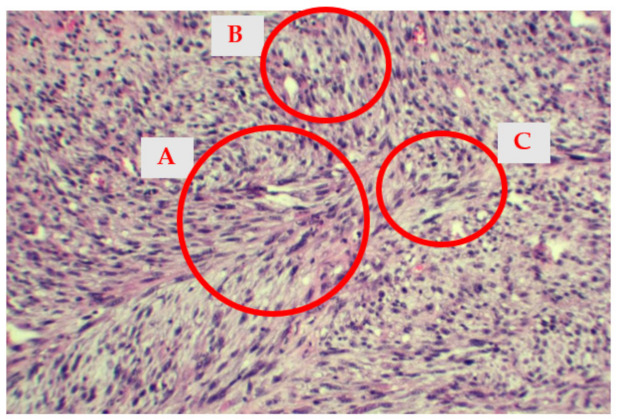
The histological appearance of the tumor showing a storiform pattern (**A**) with occasional myxoid areas (**B**). Some mitoses (**C**) are also visible (hematoxylin-eosin stain, 10× obj).

**Figure 5 medicina-58-01009-f005:**
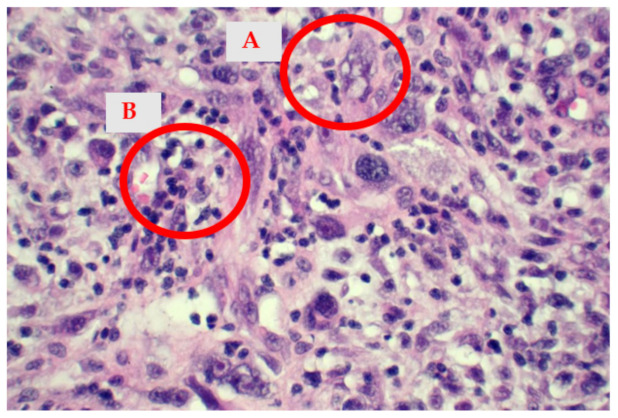
High power view of an area with marked nuclear atypia (**A**). The malignant cells have a more epithelioid appearance, however, sarcomatous morphology is still preserved. A background inflammatory infiltrate (**B**) is readily visible (hematoxylin-eosin stain, 40× obj).

## Data Availability

All the data can be found in the archive of Emergency Institute for Cardiovascular Diseases and Transplantation Targu Mures, Romania.

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
