# Peer review of "A Rare Case of Undifferentiated Pleomorphic Cardiac Sarcoma with Inflammatory Pattern"

_medicina, 2022, doi:10.3390/medicina58081009_

Round 1
Reviewer 1 Report
In this manuscript Stoica et al. report the case of 67-year-old patients with history of progressively worsening dyspnea, fatigue, intermittent fever of unknown cause and elevated markers of inflammation. An undifferentiated pleomorphic cardiac sarcoma was found at the level of the left atrium and the patient underwent cardiac surgery. After surgery the markers of inflammation decreased. However, an early recurrence occurred and the patient died in a short time.
The report well written and carries some useful teaching points.
However, some issues need to be addressed.
Major comments
-The association of UPS with fever and elevated inflammatory markers was already reported by several case reports. For this reason, the title “A curious case of…” appears inappropriate.
-Please report the value of Procalcitonin, if available. If low, this could be supportive of a paraneoplastic state rather than an infective process.
-Was transesophageal echocardiography performed before surgery?
-Please provide some echocardiographic images (transthoracic and transesophageal, if available) of the mass and the evaluation of mitral valve functional stenosis.
-Please comment also in the abstract the reasons why the “surgical margin of the resected tumor could not be evaluated” (see page 5, line 101).
Minor comments
- Page 1, line 14; typing error: “leukocytosis (26x103 u/L) 0 and high C- 14 reactive protein”.
- Page 1, line 36; “The efficacy of current treatment is unclear,”; replace “,” with “.”.
- Page 2, Line 57; Add “left” to “ventricular” and replace “contractility” with “function”.
-Page 2, line 55; left atrium is reported several times in the manuscript; abbreviate as LA.
-Page 2, Lines 59-61; Please revise the following sentence due to inconsistency: “contrast chest computed tomography […] small hypodense lesions in brain”).
-Page 3, line 84-85; improve the sentence: “According to the French Federation of Cancer Centers Sarcoma 84 Group/FNCLCC, this corresponds to score 5”.
-Page 5, line 126; “The recognition of cardiac sarcomas can be difficult, ” ; replace “,” with “.”.
- Page 5, line 127; remove “frank”.
-Page 5, line 134; remove “which can be a diagnosis dilemma in patients with malignancies. “
-Page 5, line 156; “extrapolative, chemotherapy”; replace “,” with “and”.
-Pag 5, lines 158-161; “The risk of local recurrence 158 is increased with positive surgical margins after tumor resection [25]. 159 In order to achieve complete surgical resection, UPS with inflammatory pattern 160 should be treated with neo-adjuvant chemotherapy”. These sentences are redundant, consider to remove them.
Reviewer 2 Report
I read with interest the case report "A curious case of undifferentiated pleomorphic cardiac sarcoma with inflammatory pattern" by Alexandra et. al.
The authors present a case of a 67 yo male with symptoms of cardiac congestion and fever, caused by the rare tumor of undifferentiated pleomorphic cardiac sarcoma, originating from the poster-lateral left atrium. The patient was treated by excising (most of) the tumor, however, it recurred after months with no possibility of repeat resection and the patient died later on.
There are some issues deserving attention:
- The whole course of the patient is not clear. It is at the very end that we learn that there were metastases present, also at the time of diagnosis. Please include this in the patient presentation, and the location of the metastases. So, from the beginning, this was palliative cardiac surgery intended for symptom relief, correct? Is this the reason why no neo-adjuvant therapy was used? Does this explain the "deterioration of the neurological status", i.e. brain metastases? What type of non-surgical therapy was used in detail - adjuvant chemotherapy and radiation are suggested by the narrative (Is this correct?). Please be clear on the treatment decisions taken and explain the background, so we can learn the most of it. If my assumption of palliative heart surgery is correct, a discussion of the ethics involved would also be very beneficial and a key learning point of this case.
- I think it beneficial to add "late referral" due to the unspecific symptoms and their late onset to the difficulties that need to be faced.
- I do not find the clinical presentation so "unusual". A tumor of any origin with a size of approx. 5 x 5 cm would cause dyspnea due to congestion. Esp. with the CT clearly showing the tumor occupying most of the LA, this comes as no surprise. The fatigue can also easily be explained due to the loss in stroke volume. The fever, however, is indeed not typical, as the authors point out in the discussion. But then, what would one expect of a disease that is so rare to begin with? This makes the whole case unusual, or "curious", as also already pointed out in the title.
- the Unit to the leucocytosis seems off: I guess, that "26x103 u/L" is supposed to mean "26x103 /L", still this would be very low. Did you mean "x103 /mL" or "106 /L"? Also, there is no need for the "u", since there is no "unit" but only the count of the cells.
- in the abstract, directly after the numbers for leucocytosis, there is a "0", that is not clear to me.
- Consider changing the CrP-value to 155.4 mg/L, which might be the more common unit. But I might be mistaken.
- Please consider rephrasing "functional stenosis" of the mitral valve. Since the tumor mass is occupying most of the LA, this seems not a problem of the tumor "swaying in the way" of the mitral valve during diastole, but rather a problem of insufficient preload to the LV due to the sheer volume of the tumor mass replacing LA blood. Furthermore, as the tumor is located in the LA, opening/closing of the MV should be fine, as long as there is no infiltration of the (posterior) mitral leaflet (PML). If infiltration of a leaflet is indeed the cause, as suggested by the later case unfolding (see 2.3.), then this would be a primary origin, since the leaflet itself is affected (cf. to primary and secondary mitral regurgitation). I suggest to describe what caused the stenosis (if present), maybe the PML showed reduced opening movement, suggesting tumor infiltration?
- I find it hard to believe, that the LVEF was only 50 %, if there was no dilatative cardiomyopathy of some sort: With the extend of the tumor in the LA causing massive congestion and in consequence highly reduced preload for the left ventricle (LV), I would expect a hyperkinetic LV, maybe with an LVEF of 70-80 % in order to compensate. How is this explained? Did the LVEF change after surgery?
- In figure 1, the localisation of the tumor's attachment cannot be comprehended as "inferior", since "inferior" is located towards the viewer (cf. figure 2, where the location is described correctly). What can be appreciated in the figure 1 is the tumor's (most likely) attachment to the lateral wall of the LA.
- Is there also a photograph of the actual specimen after surgical removal? This would highly enhance the case.
- There is a ")" missing at the end of figure 4. By the way: both figures 3 and 4 are of very high quality. Consider highlighting the points of interest you want the reader to appreciate, i.e. "myxoid patterns" and "mitoses" in fig. 3, and "normal cells" vs. "malignant cells", as well as the "inflammatory infiltrates" in fig. 4. - most readers will not be pathologists, I reckon.
Round 2
Reviewer 1 Report
The manuscript overall improved.
Just a few other minor comments:
-page 2, line 52: was the murmur systolic or systo-diastolic (considering the MV stenosis)?
-page 2, line 63: remove the abbreviation EF.
-page 2, line 64: replace “contrast chest and brain” with “total-body” computed tomography (also considering figures 2 and 3).
-page 2, line 79: modify as follows: “transthoracic echocardiography, apical …”
-page 3, line 82 and 87: remove the abbreviation TAP.
-page 4, figure 4 and 5, circles and letters in the figures are not clearly visible, please change the layout.
-page 5, line 124: modify as follows: “revealed a normal function of biological…”
Reviewer 2 Report
I read the revised case report "A rare case of undifferentiated pleomorphic cardiac sarcoma with inflammatory pattern" by Alexandra et. al.
The authors have greatly improved the manuscript and I think it almost ready for publication.
One thing remains: the leucocyte or white blood cell count (WBC).
Correct me if I am wrong, but my normal reference for the WBC is ~ 5-11 x 10^9/L (see also WBC count Information | Mount Sinai - New York; I am using ^9, because the program does not correctly change to "upper case"). Hence, if we are talking about leucoytosis - too many leucocytes - then 26 x 10^3/mL is somehow off. With this value, I am thinking severe leucopenia: Maybe my math is wrong, but 26 x 10^3/mL is the same as 26 x 10^6/L, thus equal to 0.026 x 10^9/L, is it not?
So, in order to have the 26 as a 2-5 fold of the normal range, either the power or the volume have to change: for instance 26 x 10^3/μL [microliter], or 26 x 10^6/mL [milliliter]. Maybe, you meant microliters, 26 x 10^3/μL? If I miscalculated, I apologize.
